# INCREMENTAL FEW-SHOT LEARNING WITH ATTENTION ATTRACTOR NETWORKS

## ABSTRACT

Machine learning classifiers are often trained to recognize a set of pre-defined classes. However, in many real applications, it is often desirable to have the flexibility of learning additional concepts, without re-training on the full training set. This paper addresses this problem, *incremental few-shot learning*, where a regular classification network has already been trained to recognize a set of base classes; and several extra novel classes are being considered, each with only a few labeled examples. After learning the novel classes, the model is then evaluated on the overall performance of both base and novel classes. To this end, we propose a meta-learning model, the Attention Attractor Network, which regularizes the learning of novel classes. In each episode, we train a set of new weights to recognize novel classes until they converge, and we show that the technique of recurrent back-propagation can back-propagate through the optimization process and facilitate the learning of the attractor network regularizer. We demonstrate that the learned attractor network can recognize novel classes while remembering old classes without the need to review the original training set, outperforming baselines that do not rely on an iterative optimization process.

## 1 INTRODUCTION

The success of deep learning stems from the availability of large scale datasets with detailed annotation, such as ImageNet (Russakovsky et al., 2015). The need for such a large dataset is however a limitation, since its collection requires intensive human labor. This is also strikingly different from human learning, where new concepts can be learned from very few examples. One line of work that attempts to bridge this gap is few-shot learning (Koch et al., 2015; Vinyals et al., 2016; Snell et al., 2017), where a model learns to output a classifier given only a few labeled examples of the unseen classes. While this is a very promising line of work, its usability in practice can be a concern, because few-shot models only focus on learning novel classes, ignoring the fact that many common classes are readily available in large datasets.

An approach that aims to enjoy the best of both worlds, the ability to learn from large datasets for common classes with the flexibility of few-shot learning for others, is *incremental few-shot learning* (Gidaris & Komodakis, 2018). This combines incremental learning where we want to add new classes without catastrophic forgetting (McCloskey & Cohen, 1989), with few-shot learning when the new classes, unlike the base classes, only have a small amount of examples. One use case to illustrate the problem is a visual aid system. Most objects of interest are common to all users, e.g., cars, pedestrian signals; however, users would also like to augment the system with additional personalized items or important landmarks in their area. Such a system needs to be able to learn new classes from few examples, without harming the performance of the original classes and without access to the original dataset used to train these classes.

In this work we present a novel method for incremental few-shot learning where during meta-learning we optimize a regularizer that reduces catastrophic forgetting from the incremental few-shot learning. Our proposed "attention attractor network" regularizer is inspired by attractor networks (Zemel & Mozer, 2001) and can be thought of as a learned memory of the base classes. We also show how this regularizer can be optimized, using recurrent back-propagation (Liao et al., 2018; Almeida, 1987; Pineda, 1987) to back-propagate through the few-shot optimization stage. Finally, we show empirically that our proposed method can produce state-of-the-art results in incremental

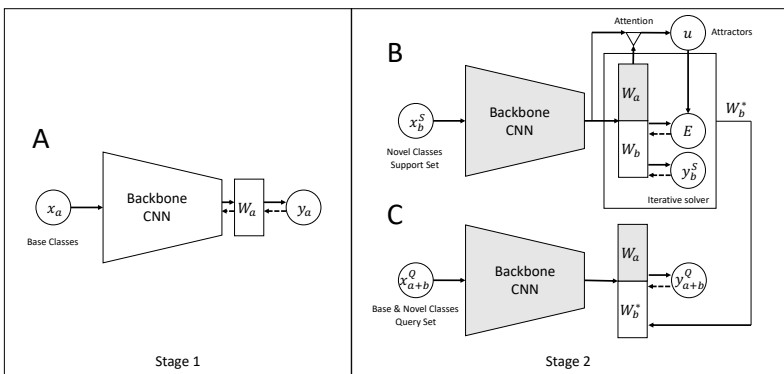

Figure 1: Our proposed attention attractor network for incremental few-shot learning. **A**: In stage 1, we learn $W_a$ and the feature extractor CNN backbone through supervised pretraining. **B**: In stage 2 we learn $W_b$ on a few-shot episode through an iterative solver, to minimize cross entropy plus an additional energy term predicted by attending to the base class representation $W_a$. **C**: The attention attractor network is learned end-to-end to minimize the expected query loss.

few-shot learning on the challenging *mini*-ImageNet (Vinyals et al., 2016) and *tiered*-ImageNet (Ren et al., 2018) tasks.

## 2 RELATED WORK

Recently, there has been a surge in interest in few-shot learning (Koch et al., 2015; Vinyals et al., 2016; Snell et al., 2017; Lake et al., 2011), where a model for novel classes is learned with only a few labeled examples. One family of approaches for few-shot learning, including Deep Siamese Networks (Koch et al., 2015), Matching Networks (Vinyals et al., 2016) and Prototypical Networks (Snell et al., 2017), follows the line of metric learning. In particular, these approaches use deep neural networks to learn a function that maps the input space to the embedding space where examples belonging to the same category are close and those belonging to different categories are far apart. Recently, Garcia & Bruna (2018) propose a graph neural networks based method which captures the information propagation from the labeled support set to the query set. Ren et al. (2018) extends Prototypical Networks to leverage unlabeled examples while doing few-shot learning. Despite the simplicity, these methods are very effective and often competitive with the state-of-the-art.

Another class of approaches try to learn models which can, unlike metric methods, adapt to the episodic tasks. In particular, Ravi & Larochelle (2017) treat the long short-term memory (LSTM) as a meta learner such that it can learn to predict the parameter update of a base learner, e.g., a convolutional neural network (CNN). MAML (Finn et al., 2017) instead learns the hyperparameters or the initial parameters of the base learner by back-propagating through the gradient descent steps. Santoro et al. (2016) use a read/write augmented memory, and Mishra et al. (2018) combine soft attention with temporal convolutions which enables retrieval of information from past episodes.

Methods described above belong to the general class of meta-learning models. First proposed in Schmidhuber (1987); Naik & Mammone (1992); Thrun (1998), meta-learning is a machine learning paradigm where the meta-learner tries to improve the base learner using the learning experiences from multiple tasks. Meta-learning methods typically learn the update policy yet lack an overall learning objective in the few-shot episodes. Furthermore, they could potentially suffer from short-horizon bias (Wu et al., 2018), if at test time the computation graph is unrolled for longer steps. To address this problem, Bertinetto et al. (2018) propose to use fast convergent models like logistic regression (LR), which can be back-propagated via a closed form update rule. Compared to Bertinetto et al. (2018), our proposed method using recurrent back-propagation (Liao et al., 2018; Almeida, 1987; Pineda, 1987) is more general as it does not require a closed-form update rule, and the inner loop solver can employ any existing continuous optimizers.

Our work is also related to incremental learning or continual learning, a setting where information is arriving continuously while prior knowledge needs to be transferred. A key challenge here is catastrophic forgetting (McCloskey & Cohen, 1989; McClelland et al., 1995), i.e., the model forgets the learned knowledge. Various forms of memory-based models have since been proposed, which either store training examples explicitly (Rebuffi et al., 2017; Sprechmann et al., 2018; Castro et al., 2018), regularize the parameter updates (Kirkpatrick et al., 2017), or learn a generative model (Kemker & Kanan, 2018). However, in these studies, incremental learning typically starts from scratch, and usually perform worse than a regular model that is trained with all available classes together since the network has not learned a good representation yet while dealing with catastophic forgetting.

To make incremental learning more useful, Hariharan & Girshick (2017); Wang et al. (2018); Gidaris & Komodakis (2018) combine few-shot learning and incremental learning, where they start off with a pre-trained network on a set of base classes, and try to augment the classifier with a batch of new classes that has not been seen during training. This is also the setting we adopt in this paper. Hariharan & Girshick (2017) propose the squared gradient magnitude loss, which makes the learned classifier from the few-shot examples have a smaller gradient value when learning on all examples. Wang et al. (2018) propose the prototypical matching networks, a combination of prototypical network and matching network. The paper also adds hallucination, which generates new examples. Both papers assume that the few-shot learning stage has access to the base classes data, which can be inflexible and inefficient due to the large size of the original training set. Gidaris & Komodakis (2018) remove this assumption and propose an attention based model which generates weights for novel categories. They also promote the use of cosine similarity between feature representations and weight vectors to classify images.

In contrast, during each few-shot episode, we directly learn a linear classifier that is solved till convergence, unlike Gidaris & Komodakis (2018) which directly output the prediction. Since in our settings the model cannot see base class data in each few-shot learning episode, different from Hariharan & Girshick (2017); Wang et al. (2018), it is challenging to jointly classify both base and novel categories using a vanilla logistic regression. Towards this end, we propose to add a learned regularizer, which is learned by differentiating through few-shot learning iterations. We found that, despite the simplicity, using an iterative solver with the learned regularizer significantly improves the linear classifier model on the task of incremental few-shot learning.

## 3 ATTENTION ATTRACTOR NETWORK FOR INCREMENTAL FEW-SHOT LEARNING

In this section, we first define the setup of incremental few-shot learning, and then we introduce our new model, the Attention Attractor Network, which attends to the set of base classes according to the few-shot training data, and learns the attractor energy function as an additional regularizing term for the few-shot episode. Figure 1 illustrates the high-level model diagram of our method.

### 3.1 INCREMENTAL FEW-SHOT LEARNING

We first define the task of incremental few-shot learning in this section. In this paper, we consider the two stage learning model proposed by Gidaris & Komodakis (2018).

**Stage 1:** We learn a base model for the regular supervised classification task on dataset $\mathcal{D}_a = \{(x_{a,i}, y_{a,i})\}_{i=1}^{N_a}$ where $x_{a,i}$ is the $i$-th example from dataset $a$ and its labeled class $y_{a,i} \in \{1, 2, ..., K\}$. The purpose of this stage is to learn both a good base classifier and a good representation. We denote parameters of the top fully connected layer as $W_a \in \mathbb{R}^{D \times K}$ where $D$ is the dimension of our learned representation.

**Stage 2:** A few-shot dataset $\mathcal{D}_b$ is presented, from which we can sample few-shot learning episodes $\mathcal{E}$. For each $N$-shot $K'$-way episode, there are $K'$ novel classes disjoint from the base classes. Each novel class has $N$ and $M$ images from the support set $S_b$ and the query set $Q_b$ respectively. Therefore, we have $\mathcal{E} = (S_b, Q_b), S_b = (x_{b,i}^S, y_{b,i}^S)_{i=1}^{N \times K'}, Q_b = (x_{b,i}^Q, y_{b,i}^Q)_{i=1}^{M \times K'}$ where $y_{b,i} \in \{K + 1, ..., K + K'\}$. $S_b$ and $Q_b$ can be regarded as the training set and the validation set in a regular supervised learning setting. To evaluate the performance on a joint prediction of both base

and novel classes, i.e., a $(K + K')$-way classification, a mini-batch $Q_a = \{(x_{a,i}, y_{a,i})\}_{i=1}^{M \times K}$ from $\mathcal{D}_a$ is also added to $Q_b$ to form $Q_{a+b} = Q_a \cup Q_b$. During this stage, the learning algorithm has only access to samples from $S_b$, i.e., few-shot examples from the new classes. But its performance is evaluated on the *joint* query set $Q_{a+b}$. For simplicity, the base model including $W_a$ is fixed so that only $W_b \in \mathbb{R}^{D \times K'}$ is learned.

In meta-training, we iteratively sample few-shot episodes $\mathcal{E}$ and try to optimize the few-shot learning algorithm, i.e., stage 2, in order to produce good results on $Q_{a+b}$. In this work, we learn a regularizer $R(\cdot; \theta_E)$ such that $W_b^* = \arg\min_{W_b} \text{Loss}(W_b, S_b) + R(W_b; \theta_E)$, and the meta-learner tries to learn $\theta_E$ such that the resulting $W_b^*$ performs well on $Q_{a+b}$.

### 3.1.1 JOINT PREDICTION ON OLD AND NEW CLASSES

When solving a few-shot episode, we consider learning a linear classifier on the novel classes in the support set $S_b$. For notational convenience, we augment the hidden representation $h$ as below such that the 0-th dimension of the weights is the original bias.

$$h = [1, \tilde{h}], \quad W^\top h = W_{1:}^\top \tilde{h} + W_0. \tag{1}$$

We use the concatenation of $W_a$ and $W_b$ as the fully-connected layer for classification,

$$\hat{y}_{b,i} = \text{softmax}([W_a, W_b]^\top h(x_{b,i})). \tag{2}$$

Here we emphasize that learnable parameters of $\hat{y}_{b,i}$ is $W_b$. During the learning stage of each few-shot episode, we treat it as a classification problem and aim to minimize the following regularized cross-entropy objective on the support set $S$,

$$W_b^* = \arg\min L^S(W_b) = \frac{1}{NK'} \sum_{i=1}^{NK'} \sum_{c=1}^{K+K'} y_{b,i,c}^S \log \hat{y}_{b,i,c}^S(W_b) + R(W_b; \theta_E) \tag{3}$$

### 3.2 ATTENTION ATTRACTOR NETWORKS

Directly learning the few-shot episode, e.g., by setting $R(W_b; \theta_E)$ to be zero or simple weight decay, can cause catastrophic forgetting on the old classes. This is because $W_b$ is trying to minimize the probability of all the old classes, as it is trained only on new classes. In this section, we introduce Attention Attractor Networks to address this problem.

The attractor network adds an energy term $E$ to the few-shot learning objective, $R(W_b; \theta_E) = \frac{1}{2}\lambda\|W_b\|_2^2 + E(W_b; \theta_E)$.

$$W_b^*(\theta_E, S_b) = \arg\min L^S(W_b; \theta_E) \tag{4}$$

$$= \arg\min \frac{1}{NK'} \sum_{i=1}^{NK'} \sum_{c=1}^{K+K'} y_{b,i,c} \log \hat{y}_{b,i,c} + \frac{1}{2}\lambda\|W_b\|_2^2 + E(W_b; \theta_E), \tag{5}$$

where the optimized parameters $W_b^*$ are functions of energy function parameters $\theta_E$ and few-shot samples $S_b$. During meta-learning, $\theta_E$ are updated to minimize an expected loss of the query set $Q_{a+b}$ which contains both old and new classes, averaging over all few-shot learning episodes.

$$\hat{y}_j(\theta_E, S_b) = \text{softmax}\left([W_a, W_b^*(\theta_E, S_b)]^\top h(x_j)\right), \tag{6}$$

$$\theta_E^* = \arg\min_{\theta_E} \mathbb{E}_{\mathcal{E}}[L^Q(\theta_E, S_b)] = \arg\min_{\theta_E} \mathbb{E}_{\mathcal{E}} \left[ \sum_{j=1}^{M(K+K')} \sum_{c=1}^{K+K'} y_{j,c} \log \hat{y}_{j,c}(\theta_E, S_b) \right]. \tag{7}$$

Conceptually, the energy function regularizes the learning of $W_b$ so that it is compatible with $W_a$ during the joint prediction. In our proposed model each base class in $W_a$ has a learned attractor $U_k$ stored in the knowledge base matrix $U = [U_1, ..., U_K]$. When a novel class $k'$ is seen, its classifier is regularized towards its attractor $u_{k'}$ which is a weighted sum of $U_k$. The weighting can be seen

as an attention mechanism where each new class attends to the old classes according to the level of interference. Specifically, the attention mechanism is implemented by the cosine similarity function:

$$A(x, y) = \frac{x^\top y}{\|x\|_2 \|y\|_2}. \tag{8}$$

For each image in the support set, we compare it with the set of base weights $W_a$, average over each support class and apply a softmax function,

$$a_{k',k} = \frac{\exp\left(\frac{1}{N} \sum_j \tau A(h_j, W_{a,k}) \mathbb{1}[y_{b,j} = k']\right)}{\sum_k \exp\left(\frac{1}{N} \sum_j \tau A(h_j, W_{a,k}) \mathbb{1}[y_{b,j} = k']\right)}, \tag{9}$$

where $h_j$ are the representations of the inputs in the support set $S_b$ and $\tau$ is a learnable temperature scalar. $a_{k',k}$ encodes a normalized pairwise attention matrix between the novel classes and the base classes. The attention vector is then used to compute a linear weighted sum of entries in the knowledge base matrix $U$,

$$u_{k'} = \sum_k a_{k',k} U_k + U_0. \tag{10}$$

The final energy function is defined as a sum of squared $L_2$ distance from the attractors,

$$E(W_b; \theta_E) = \frac{1}{2} \sum_{k'} (W_{b,k'} - u_{k'})^\top \mathrm{diag}(\exp(\gamma))(W_{b,k'} - u_{k'}), \tag{11}$$

where $\gamma$ is a learnable vector, which defines a diagonal distance metric over the hidden dimensions. Our proposed attention attractor network takes inspiration from attractor networks (Mozer, 2009; Zemel & Mozer, 2001), where for each base class we learn an "attractor" that stores the relevant memory regrading that class.

In summary, $\theta_E$ is a tuple of $(U, U_0, \gamma, \tau)$. The number of parameters is on the same order as a fully connected layer $W_a$. It is important to note that $E(W_b; \theta_E)$ is convex in $W_b$, so the optimum $W_b^*(\theta_E, S_b)$ is guaranteed to be unique and achievable.

### 3.3 LEARNING AN ENERGY FUNCTION USING RECURRENT BACKPROPAGATION

As there is no closed-form solution to the regularized linear classification problem shown above, in each training step, we need to solve for $L^S$ to obtain $W_b^*$ through an iterative optimizer. The question is how to efficiently compute $\frac{\partial W_b^*}{\partial \theta_E}$, back-propagating through the optimization. For simple energy functions, we can unroll the iterative optimization process in the computation graph and use the backpropagation through time (BPTT) algorithm (Werbos, 1990). However, the number of iterations for a gradient-based optimizer to converge can be on the order of thousands, and BPTT can be computationally prohibitive. Another way is to use the truncated BPTT (Williams & Peng, 1990) (T-BPTT) algorithm that optimizes for the initial $T$ steps of gradient-based optimization, and is commonly used in meta-learning problems. However, when $T$ is small the training objective is biased compared to the objective that considers the optimal $W_b$.

Alternatively, the recurrent backpropagation algorithm (Almeida, 1987; Pineda, 1987; Liao et al., 2018) allows us to backpropagate through fixed-point iterations efficiently without unrolling the computation graph and storing intermediate activations. Consider a vanilla gradient descent process on $W_b$ with step size $\alpha$. The difference between two steps $\Phi$ can be written as $\Phi(W_b^{(t)}) = W_b^{(t)} - F(W_b^{(t)})$, where $F(W_b^{(t)}) = W_b^{(t+1)} = W_b^{(t)} - \alpha \nabla L^S(W_b^{(t)})$. Since $\Phi(W_b^*(\theta_E))$ is identically zero as a function of $\theta_E$, we have

$$\frac{\partial W_b^*}{\partial \theta_E} = (I - J_{F,W_b^*}^\top)^{-1} \frac{\partial F}{\partial \theta_E}, \tag{12}$$

where $J_{F,W_b^*}$ denotes the Jacobian matrix of the mapping $F$ evaluated at $W_b^*$.

**Damped Neumann RBP**  To compute the matrix-inverse vector product $(I - J^\top)^{-1} v$ in Equation 12, Liao et al. (2018) propose to use the Neumann series:

$$(I - J^\top)^{-1} v = \sum_{n=0}^{\infty} (J^\top)^n v \equiv \sum_{n=0}^{\infty} v^{(n)}. \tag{13}$$

---

**Algorithm 1** Learning an Energy Function for Incremental Few-Shot Learning

---

**Require:** $\theta_0, \mathcal{D}_a, \mathcal{D}_b, N, K', h$
**Ensure:** $\theta_E$
1: $\theta_E \leftarrow \theta_0$;
2: **for** $t = 1 \dots T$ **do**
3: $\quad \{(x_b^S, y_b^S)\}, \{(x_b^Q, y_b^Q)\} \leftarrow \text{SampleEpisode}(\mathcal{D}_b, N, K')$;
4: $\quad \{x_{a+b}^Q, y_{a+b}^Q\} \leftarrow \text{SampleMiniBatch}(\mathcal{D}_a, NK') \cup \{(x_b^Q, y_b^Q)\}$;
5: $\quad$ **repeat**
6: $\quad\quad \hat{y}_{b,j}^S \leftarrow \text{softmax}([W_a, W_b]^\top h(x_{b,j}^S))$;
7: $\quad\quad L^S \leftarrow \frac{1}{NK'} \sum_i y_{b,i}^S \log \hat{y}_{b,i}^S + \lambda \|W_b\|_2^2 + E(W_b; \theta_E)$;
8: $\quad\quad W_b \leftarrow \text{OptimizerStep}(W_b, \nabla_{W_b} L^S)$;
9: $\quad$ **until** $W_b$ converges
10: $\quad \hat{y}_{a+b,j}^Q \leftarrow \text{softmax}([W_a, W_b]^\top h(x_{a+b,j}^Q)), \ \forall j$;
11: $\quad L^Q \leftarrow \frac{1}{2NK'} \sum_j y_{a+b,j}^Q \log \hat{y}_{a+b,j}^Q$;
$\quad\quad$ // Backprop through the above converging process
$\quad\quad$ // A dummy gradient descent step
12: $\quad W_b' \leftarrow W_b - \alpha \nabla_{W_b} L^S$;
13: $\quad J \leftarrow \frac{\partial W_b'}{\partial W_b}; v \leftarrow \frac{\partial L^Q}{\partial W_b}; g \leftarrow v$;
14: $\quad$ **repeat**
15: $\quad\quad v \leftarrow J^\top v - \epsilon v; g \leftarrow g + v$;
16: $\quad$ **until** $g$ converges
17: $\quad \theta_E \leftarrow \text{OptimizerStep}(\theta_E, g^\top \frac{\partial W_b}{\partial \theta_E})$
18: **end for**

---

Note that the matrix vector product $J^\top v$ can be computed by standard backpropagation. Nonetheless, we empirically observe that directly applying the Neumann RBP algorithm leads to numerical instability. Therefore, we propose "damped Neumann RBP", which adds a damping term $0 < \epsilon < 1$ to $I - J^\top$. It results in the following update: $\tilde{v}^{(n)} = (J^\top - \epsilon I)^n v$. The damping constant $\epsilon$ can also be interpreted as an additional weight decay term of the energy function during the backpropagation steps $E^B(W_b) = E(W_b) + \frac{\epsilon}{2\alpha} \|W_b\|_2^2$. In practice, we found the damping term with $\epsilon = 0.1$ helps alleviate the issue significantly.

Algorithm 1 outlines the key steps for learning an energy function using RBP in the incremental few-shot learning setting.

## 4 EXPERIMENTS

### 4.1 DATASETS

We used few-shot image classification benchmarks, *mini*-ImageNet Vinyals et al. (2016) and *tiered*-ImageNet Ren et al. (2018), and made modifications of the datasets to accommodate the incremental few-shot learning settings. For meta-learning we need to split between training base classes, and training classes from which we sample few-shot episodes. In *tiered*-ImageNet we split the 351 training classes to 200 base classes and 151 novel classes used to train the meta-learner. In *mini*-ImageNet this is problematic as there are only 64 training classes, therefore at each meta-learning episode we randomly erase 5 classes, considering the 59 left as our base classes and sampling a few-shot episode from the removed 5 classes. For validation/test we sample from the validation/test set of the base classes and sample episodes from separate validation and test classes. Further details about these datasets and the class splits are in the supplementary material.

### 4.2 EXPERIMENTAL SETUP

In the first stage, we use a standard ResNet backbone (He et al., 2016) to learn the feature representation through supervised training. For *mini*-ImageNet experiments, we follow Mishra et al. (2018) and use a modified version of ResNet. For *tiered*-ImageNet, we use the standard ResNet-18 (He et al., 2016), but replace all batch normalization (Ioffe & Szegedy, 2015) layers with group normalization (Wu & He, 2018), as there is a large distributional shift from training to testing in

Table 1: *mini-ImageNet* 64+5-way few-shot classification results

| Model | Backbone | Base | 1-shot | | | 5-shot | | |
| --- | --- | --- | --- | --- | --- | --- | --- | --- |
| | | | Novel | Both | $\overline{\Delta}$ | Novel | Both | $\overline{\Delta}$ |
| MatchingNets (Vinyals et al., 2016) | C64 | - | 43.60 | - | - | 55.30 | - | - |
| Meta-LSTM (Ravi & Larochelle, 2017) | C32 | - | $43.40 \pm 0.77$ | - | - | $60.20 \pm 0.71$ | - | - |
| MAML (Finn et al., 2017) | C64 | - | $48.70 \pm 1.84$ | - | - | $63.10 \pm 0.92$ | - | - |
| RelationNet (Sung et al., 2018) | C64 | - | $50.44 \pm 0.82$ | - | - | $65.32 \pm 0.70$ | - | - |
| R2-D2 (Bertinetto et al., 2018) | C256 | - | $51.20 \pm 0.60$ | - | - | $68.20 \pm 0.60$ | - | - |
| SNAIL (Mishra et al., 2018) | ResNet | - | $55.71 \pm 0.99$ | - | - | $68.88 \pm 0.92$ | - | - |
| ProtoNet (Snell et al., 2017) | C64 | - | $49.42 \pm 0.78$ | - | - | $68.20 \pm 0.66$ | - | - |
| ProtoNet (our implementation) | ResNet | 75.79 | $50.09 \pm 0.41$ | 42.73 | -20.21 | $70.76 \pm 0.19$ | 57.05 | -31.72 |
| LwoF (Gidaris & Komodakis, 2018) | ResNet | 80.24 | $55.45 \pm 0.89$ | 51.23 | - | $70.92 \pm 0.35$ | 56.04 | - |
| LwoF (our implementation) | ResNet | 74.58 | $56.97 \pm 0.24$ | 52.37 | -13.65 | $70.50 \pm 0.36$ | 59.90 | -14.18 |
| Ours (1st stage) | ResNet | 77.17 | $54.78 \pm 0.43$ | 52.74 | -13.95 | $70.57 \pm 0.36$ | 60.34 | -13.60 |
| Ours (full model) | ResNet | 76.84 | $55.72 \pm 0.41$ | **54.89** | **-11.39** | $70.50 \pm 0.36$ | **62.37** | **-11.48** |

Table 2: *tiered-ImageNet* 200+5-way few-shot classification results

| Model | Backbone | Base | 1-shot | | | 5-shot | | |
| --- | --- | --- | --- | --- | --- | --- | --- | --- |
| | | | Novel | Both | $\overline{\Delta}$ | Novel | Both | $\overline{\Delta}$ |
| ProtoNet (Snell et al., 2017) | ResNet | 59.70 | $48.19 \pm 0.43$ | 34.49 | -19.45 | $65.90 \pm 0.19$ | 50.27 | -12.54 |
| LwoF (Gidaris & Komodakis, 2018) | ResNet | 61.84 | $50.90 \pm 0.46$ | 54.05 | -2.35 | $66.69 \pm 0.36$ | 62.32 | -1.90 |
| Ours (1st stage) | ResNet | 62.01 | $47.09 \pm 0.42$ | 48.58 | -5.95 | $64.90 \pm 0.41$ | 59.73 | -3.72 |
| Ours (full model) | ResNet | 61.59 | $51.12 \pm 0.45$ | **55.56** | **-0.80** | $66.40 \pm 0.36$ | **63.27** | **-0.83** |

*tiered*-ImageNet due to categorical splits. We used standard data augmentation, with random crops and random horizonal flips. All of our models in our implementations use the same pretrained model as the starting point for the second stage.

In the second stage of learning as well as the final evaluation, we sample a few-shot episode from the $\mathcal{D}_b$, together with a regular mini-batch from the $\mathcal{D}_a$. The base class images are added to the query set of the few-shot episode. The base and novel classes are maintained in equal proportion in our experiments. For all the experiments, we consider 5-way classification with 1 or 5 support examples (i.e. shots); the total number of query images per episode is $75 \times 2 = 150$.

We use an L-BFGS (Zhu et al., 1997) to solve the inner loop of our models to make sure $W_b$ converges. For BPTT baselines, we unroll the computation graph using vanilla gradient descent with a fixed learning rate 1e-2. We use the ADAM (Kingma & Ba, 2015) optimizer for meta-learning with a learning rate of 1e-3, which decays by a factor of 10 after 4,000 steps, for a total of 8,000 steps. We fix the RBP to 20 iterations and $\epsilon = 0.1$.

### 4.3 EVALUATION METRICS

We consider the following evaluation metrics: 1) overall accuracy on individual query sets and the joint query set ("Base", "Novel", and "Both"); and 2) decrease in performance during joint prediction within the base and novel classes, considered separately ("$\Delta_a$" and "$\Delta_b$"). Finally we take the average $\overline{\Delta} = \frac{1}{2}(\Delta_a + \Delta_b)$ as a key measure of the overall decrease in accuracy.

### 4.4 RESULTS

We present the few-shot benchmark in Table 1 and 2. On *mini*-ImageNet, we compare our models to other incremental few-shot learning methods as well as pure few-shot learning methods. Note that our model uses the same backbone architecture as Mishra et al. (2018) and Gidaris & Komodakis (2018), and is directly comparable with their results. The "1st stage" baseline is a CNN trained on the base classes, where we train the new classes using logistic regression with weight decay.

In the incremental few-shot learning setting, we implemented and compared to two methods. First, we adapted ProtoNet (Snell et al., 2017) to incremental few-shot settings. For each base class we store a base representation, which is the average representation over all images belonging to the base class. During the few-shot learning stage, we again average the representation of the few-shot classes and add them to the bank of base representations. Finally, we retrieve the nearest neighbor by comparing the representation of a test image with entries in the representation store. In summary, both $W_a$ and $W_b$ are stored as the average representation of all images seen so far that belong to a certain class. We also compare to Dynamic Few-Shot Learning without Forgetting

Table 3: Learning an attention attractor network using damped Neumann RBP vs. truncated BPTT. Models are on evaluated on the validation set of *tiered*-ImageNet. During evaluation, "*" models are solved till convergence using a second-order optimizer.

| 1-shot | Base | Base+ | $\Delta_a$ | Novel | Novel+ | $\Delta_b$ | Both | $\overline{\Delta}$ |
|---|---|---|---|---|---|---|---|---|
| BPTT 20 Steps | 61.41 | 61.16 | -0.25 | $42.09 \pm 0.40$ | $37.80 \pm 0.37$ | -4.28 | 49.44 | -2.27 |
| BPTT 100 Steps | 61.41 | 61.08 | -0.33 | $44.49 \pm 0.42$ | $40.43 \pm 0.38$ | -4.06 | 50.75 | -2.20 |
| BPTT 200 Steps | 61.43 | 60.86 | -0.57 | $45.49 \pm 0.42$ | $42.40 \pm 0.40$ | -3.09 | 51.63 | -1.83 |
| BPTT 20 Steps * | 61.65 | 6.02 | -55.63 | $27.08 \pm 0.26$ | $21.99 \pm 0.24$ | -5.08 | 14.01 | -30.36 |
| BPTT 100 Steps * | 61.46 | 5.23 | -56.22 | $27.70 \pm 0.29$ | $19.59 \pm 0.26$ | -8.11 | 12.41 | -32.17 |
| BPTT 200 Steps * | 61.52 | 5.62 | -55.89 | $29.31 \pm 0.32$ | $21.09 \pm 0.30$ | -8.22 | 13.36 | -32.06 |
| Ours (RBP 20 Steps) | 61.52 | 61.11 | -0.41 | $48.72 \pm 0.45$ | $47.04 \pm 0.45$ | -1.68 | **54.08** | **-1.05** |

Table 4: Ablation studies on *tiered*-ImageNet. "Base+" and "Novel+" are the prediction accuracies on Base and Novel classes within a joint query set.

| 1-shot | Base | Base+ | $\Delta_a$ | Novel | Novel+ | $\Delta_b$ | Both | $\overline{\Delta}$ |
|---|---|---|---|---|---|---|---|---|
| None | 61.63 | 59.61 | -2.02 | $43.68 \pm 0.40$ | $34.59 \pm 0.34$ | -9.10 | 47.10 | -5.56 |
| Best WD ($\lambda$=0) | 61.63 | 59.61 | -2.02 | $43.68 \pm 0.40$ | $34.59 \pm 0.34$ | -9.10 | 47.10 | -5.56 |
| Gamma Random | 61.84 | 61.28 | -0.56 | $44.29 \pm 0.40$ | $18.66 \pm 0.29$ | -25.63 | 39.97 | -13.09 |
| Gamma w/o $W_a$ | 61.97 | 61.57 | -0.40 | $48.05 \pm 0.43$ | $32.16 \pm 0.38$ | -15.89 | 46.86 | -8.14 |
| Gamma | 61.68 | 60.43 | -1.25 | $46.17 \pm 0.42$ | $43.39 \pm 0.41$ | -2.77 | 51.91 | -2.01 |
| Fixed Attractor | 61.74 | 61.06 | -0.67 | $45.96 \pm 0.42$ | $44.30 \pm 0.42$ | -1.66 | 52.68 | -1.17 |
| Full Model | 61.52 | 61.11 | -0.41 | $48.72 \pm 0.45$ | $47.04 \pm 0.45$ | -1.68 | **54.08** | **-1.05** |
| 5-shot | Base | Base+ | $\Delta_a$ | Novel | Novel+ | $\Delta_b$ | Both | $\overline{\Delta}$ |
| None | 61.75 | 52.74 | -9.01 | $61.00 \pm 0.39$ | $59.82 \pm 0.38$ | -1.18 | 56.28 | -5.09 |
| Best WD ($\lambda$=5e-4) | 61.62 | 58.48 | -3.14 | $61.59 \pm 0.39$ | $57.93 \pm 0.37$ | -3.65 | 58.21 | -3.39 |
| Gamma Random | 61.78 | 59.42 | -2.36 | $61.20 \pm 0.38$ | $55.66 \pm 0.36$ | -5.54 | 57.54 | -3.95 |
| Gamma w/o $W_a$ | 62.00 | 57.99 | -4.01 | $61.18 \pm 0.40$ | $59.00 \pm 0.39$ | -2.18 | 58.49 | -3.09 |
| Gamma | 61.46 | 59.87 | -1.59 | $61.72 \pm 0.39$ | $59.80 \pm 0.39$ | -1.92 | 59.83 | -1.76 |
| Fixed Attractor | 61.78 | 60.82 | -0.97 | $61.33 \pm 0.39$ | $60.06 \pm 0.39$ | -1.28 | 60.44 | -1.12 |
| Full Model | 61.80 | 61.28 | -0.53 | $63.40 \pm 0.39$ | $62.09 \pm 0.39$ | -1.31 | **61.68** | **-0.92** |

(LwoF) (Gidaris & Komodakis, 2018). Here the base weights $W_a$ are learned regularly through supervised pre-training, and $W_b$ are computed using prototypical averaging. We implemented the most advanced variants proposed in the paper, which involves a class-wise attention mechanism. The main difference to this work is that we use an iterative optimization to compute $W_b$.

Shown in Table 1 and 2, our proposed method consistently outperform the two approaches described above, particularly in "$\overline{\Delta}$". Our method is significantly improved during the second stage training, compared to a baseline that has only been pre-trained on base classes in the first stage.

### 4.5 COMPARISON TO TRUNCATED BPTT

An alternative way to learn the energy function is to unroll the inner optimization for a fixed number of steps in a differentiable computation graph, and then we can back-propagate through time. In fact, truncated BPTT is a popular learning algorithm in recent meta-learning approaches (Andrychowicz et al., 2016; Ravi & Larochelle, 2017; Finn et al., 2017; Sprechmann et al., 2018).

In Table 6, we report the performance of the truncated BPTT versions of our attention attractor model on 1-shot learning. Results on 5-shot are in the supplementary material. We unrolled the computation graph by 20 or 100 steps. In comparison, the RBP version also runs for only 20 truncation Neumann steps. At test time, we also test the BPTT learned model by solving the learned energy function until convergence, just like the RBP model. We can see that truncated BPTT versions have worse performance than RBP, since solving the fully connected layer weights requires more iterations than the number of unrolling steps. When we tried to solve the optimization to convergence at test time, we found that the BPTT model performance greatly decayed (colored in red), as they are only guaranteed to work well for a certain number of steps, and failed to learn a good energy function. In contrast, learning the energy function with RBP finds a good solution upon convergence, regardless of the optimizer being used.

## 4.6 ABLATION STUDIES

To understand the effectiveness of each part of the proposed model, we consider the following baselines as control experiments:

- **Plain logistic regression** ("None") solves a logistic regression problem at each few-shot episode, without any form of weight decay other regularizers.
- **Scalar weight decay** ("WD") has a constant $\lambda$ for weight decay as the regularizer. The $\lambda$ is chosen according to validation performance.
- **Learnable** $\gamma$ ("Gamma") learns $\gamma$, which affects the size of the attractor basin, while keeping the center of the attractor $u$ at zero. "w/o $W_a$" variant does not fine-tune $W_a$ in the second stage.
- **Fixed attractor** learns a fixed attractor center, in addition to learning $\gamma$. No attention mechanism is employed here.

Experimental results are reported in Table 4. While models have similar performance on individually predicting base and novel classes, since they are using the same pre-trained backbone, there are significant differences in terms of joint prediction. Adding $\gamma$, fine-tuning $W_a$, adding attractors, and using our attention mechanism all contribute towards the final performance. In particular, on $\overline{\Delta}$, the full attention attractor model is 80% better compared to no regularization, and 10-18% relative to to the fixed attractor.

## 5 CONCLUSION AND FUTURE WORK

Incremental few-shot learning, the ability to jointly predict based on a set of pre-defined concepts as well as additional novel concepts, is an important step towards making machine learning models more flexible and usable in everyday life. In this work, we propose an attention attractor model, which outputs an additional energy function by attending to the set of base classes. We show that recurrent back-propagation is an effective and modular tool for learning energy functions in a general meta-learning setting, whereas truncated back-propagation through time fails to learn functions that converge well. Our attention attractor network is an iterative model that learns to remember the base classes without needing to review examples from the original training set, outperforming baselines that only do one-step inference. Future directions of this work include sequential iterative learning of few-shot novel concepts, and hierarchical memory organization.

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

# A  DATASETS

We experiment on two datasets, *mini*-ImageNet and *tiered*-ImageNet. Both are sub-sets of imagenet (Russakovsky et al., 2015), with images size reduced to size of $84 \times 84$ pixels

- ***mini*-ImageNet** Proposed by Vinyals et al. (2016), *mini*-ImageNet which contains 100 object classes and 60,000 images. We used the splits proposed by Ravi & Larochelle (2017), where training, validation, and testing have 64, 16, and 20 classes respectively.

- ***tiered*-ImageNet** Proposed by Ren et al. (2018), *tiered*-ImageNet is a larger subset of ILSVRC-12. It features a categorical split among training, validation, and testing subsets. The categorical split means that classes that belong to the same high-level category, e.g. "working dog", are not split between training, validation and test. This is harder task, but one that more truthfully evaluates generalization to new classes. It is also an order of magnitude larger than *mini*-ImageNet.

Details about number of samples are presented in Table 5.

Table 5: *mini*-ImageNet and *tiered*-ImageNet split statistics

| Classes | Purpose | *mini*-ImageNet | | | *tiered*-ImageNet | | |
| --- | --- | --- | --- | --- | --- | --- | --- |
| | | Split | N. Cls | N. Img | Split | N. Cls | N. Img |
| Base | Train | Train-Train | 64 | 38,400 | Train-A-Train | 200 | 203,751 |
| | Val | Train-Val | 64 | 18,748 | Train-A-Val | 200 | 25,460 |
| | Test | Train-Test | 64 | 19,200 | Train-A-Test | 200 | 25,488 |
| Novel | Train | Train-Train | 64 | 38,400 | Train-B | 151 | 193,996 |
| | Val | Val | 16 | 9,600 | Val | 97 | 124,261 |
| | Test | Test | 20 | 12,000 | Test | 160 | 206,209 |

## A.1  VALIDATION AND TESTING SPLITS FOR BASE CLASSES

In standard few-shot learning, meta-training, validation, and test set have disjoint sets of object classes. However, in our incremental few-shot learning setting, to evaluate the model performance on the base class predictions, additional splits of validation and test splits of the meta-training set are required. Splits and dataset statistics are listed in Table 5. For *mini*-ImageNet, Gidaris & Komodakis (2018) released additional images for evaluating training set, namely "Train-Val" and "Train-Test". For *tiered*-ImageNet, we split out $\approx 20\%$ of the images for validation and testing of the base classes.

## A.2  NOVEL CLASSES

In *mini*-ImageNet experiments, the same training set is used for both $\mathcal{D}_a$ and $\mathcal{D}_b$. In order to pretend that the classes in the few-shot episode are novel, following Gidaris & Komodakis (2018), we masked the base classes in $W_a$, which contains 64 base classes. In other words, we essentially train for a 59+5 classification task. We found that under this setting, the progress of meta-learning in the second stage is not very significant, since all classes have already been seen before.

In *tiered*-ImageNet experiments, to emulate the process of learning novel classes during the second stage, we split the training classes into base classes ("Train-A") with 200 classes and novel classes ("Train-B") with 151 classes, just for meta-learning purpose. During the first stage the classifier is trained using Train-A-Train data. In each meta-learning episode we sample few-shot examples from the novel classes (Train-B) and a query base set from Train-A-Val.

**200 Base Classes ("Train-A"):**

```
n02128757, n02950826, n01694178, n01582220, n03075370, n01531178,
n03947888, n03884397, n02883205, n03788195, n04141975, n02992529,
n03954731, n03661043, n04606251, n03344393, n01847000, n03032252,
n02128385, n04443257, n03394916, n01592084, n02398521, n01748264,
n04355338, n02481823, n03146219, n02963159, n02123597, n01675722,
n03637318, n04136333, n02002556, n02408429, n02415577, n02787622,
```

```
n04008634, n02091831, n02488702, n04515003, n04370456, n02093256,
n01693334, n02088466, n03495258, n02865351, n01688243, n02093428,
n02410509, n02487347, n03249569, n03866082, n04479046, n02093754,
n01687978, n04350905, n02488291, n02804610, n02094433, n03481172,
n01689811, n04423845, n03476684, n04536866, n01751748, n02028035,
n03770439, n04417672, n02988304, n03673027, n02492660, n03840681,
n02011460, n03272010, n02089078, n03109150, n03424325, n02002724,
n03857828, n02007558, n02096051, n01601694, n04273569, n02018207,
n01756291, n04208210, n03447447, n02091467, n02089867, n02089973,
n03777754, n04392985, n02125311, n02676566, n02092002, n02051845,
n04153751, n02097209, n04376876, n02097298, n04371430, n03461385,
n04540053, n04552348, n02097047, n02494079, n03457902, n02403003,
n03781244, n02895154, n02422699, n04254680, n02672831, n02483362,
n02690373, n02092339, n02879718, n02776631, n04141076, n03710721,
n03658185, n01728920, n02009229, n03929855, n03721384, n03773504,
n03649909, n04523525, n02088632, n04347754, n02058221, n02091635,
n02094258, n01695060, n02486410, n03017168, n02910353, n03594734,
n02095570, n03706229, n02791270, n02127052, n02009912, n03467068,
n02094114, n03782006, n01558993, n03841143, n02825657, n03110669,
n03877845, n02128925, n02091032, n03595614, n01735189, n04081281,
n04328186, n03494278, n02841315, n03854065, n03498962, n04141327,
n02951585, n02397096, n02123045, n02095889, n01532829, n02981792,
n02097130, n04317175, n04311174, n03372029, n04229816, n02802426,
n03980874, n02486261, n02006656, n02025239, n03967562, n03089624,
n02129165, n01753488, n02124075, n02500267, n03544143, n02687172,
n02391049, n02412080, n04118776, n03838899, n01580077, n04589890,
n03188531, n03874599, n02843684, n02489166, n01855672, n04483307,
n02096177, n02088364.
```

**151 Novel Classes ("Train-B"):**

```
n03720891, n02090379, n03134739, n03584254, n02859443, n03617480,
n01677366, n02490219, n02749479, n04044716, n03942813, n02692877,
n01534433, n02708093, n03804744, n04162706, n04590129, n04356056,
n01729322, n02091134, n03788365, n01739381, n02727426, n02396427,
n03527444, n01682714, n03630383, n04591157, n02871525, n02096585,
n02093991, n02013706, n04200800, n04090263, n02493793, n03529860,
n02088238, n02992211, n03657121, n02492035, n03662601, n04127249,
n03197337, n02056570, n04005630, n01537544, n02422106, n02130308,
n03187595, n03028079, n02098413, n02098105, n02480855, n02437616,
n02123159, n03803284, n02090622, n02012849, n01744401, n06785654,
n04192698, n02027492, n02129604, n02090721, n02395406, n02794156,
n01860187, n01740131, n02097658, n03220513, n04462240, n01737021,
n04346328, n04487394, n03627232, n04023962, n03598930, n03000247,
n04009552, n02123394, n01729977, n02037110, n01734418, n02417914,
n02979186, n01530575, n03534580, n03447721, n04118538, n02951358,
n01749939, n02033041, n04548280, n01755581, n03208938, n04154565,
n02927161, n02484975, n03445777, n02840245, n02837789, n02437312,
n04266014, n03347037, n04612504, n02497673, n03085013, n02098286,
n03692522, n04147183, n01728572, n02483708, n04435653, n02480495,
n01742172, n03452741, n03956157, n02667093, n04409515, n02096437,
n01685808, n02799071, n02095314, n04325704, n02793495, n03891332,
n02782093, n02018795, n03041632, n02097474, n03404251, n01560419,
n02093647, n03196217, n03325584, n02493509, n04507155, n03970156,
n02088094, n01692333, n01855032, n02017213, n02423022, n03095699,
n04086273, n02096294, n03902125, n02892767, n02091244, n02093859,
n02389026.
```

# B  5-SHOT RESULTS

We show here additional results on 5-shot learning. In table **??** we compare RBP to BPTT optimization.

Table 6: Learning an attention attractor network using damped Neumann RBP vs. truncated BPTT. Models are on evaluated on the validation set of *tiered*-ImageNet. During evaluation, "*" models are solved till convergence using a second-order optimizer.

| 5-shot | Base | Base+ | $\Delta_a$ | Novel | Novel+ | $\Delta_b$ | Both | $\overline{\Delta}$ |
|---|---|---|---|---|---|---|---|---|
| BPTT 20 Steps | 61.48 | 61.05 | -0.43 | $47.72 \pm 0.37$ | $41.72 \pm 0.37$ | -6.00 | 51.39 | -3.21 |
| BPTT 100 Steps | 61.75 | 61.28 | -0.47 | $59.51 \pm 0.37$ | $55.79 \pm 0.36$ | -3.72 | 58.53 | -2.10 |
| BPTT 200 Steps | 61.59 | 60.79 | -0.79 | $62.50 \pm 0.39$ | $60.40 \pm 0.38$ | -2.10 | 60.60 | -1.44 |
| BPTT 20 Steps * | 61.49 | 7.29 | -54.20 | $37.16 \pm 0.33$ | $37.11 \pm 0.33$ | -0.05 | 22.20 | -27.12 |
| BPTT 100 Steps * | 61.46 | 4.00 | -57.45 | $41.68 \pm 0.36$ | $41.67 \pm 0.36$ | 0.00 | 22.84 | -28.73 |
| BPTT 200 Steps * | 61.37 | 5.63 | -55.74 | $43.74 \pm 0.37$ | $43.73 \pm 0.36$ | 0.00 | 24.68 | -27.87 |
| Ours (RBP 20 Steps) | 61.80 | 61.28 | -0.53 | $63.40 \pm 0.39$ | $62.09 \pm 0.39$ | -1.31 | **61.68** | **-0.92** |

