# OpenReview forum: "Incremental Few-Shot Learning with Attention Attractor Networks"
_ICLR.cc/2019/Conference_

### Official Review · AnonReviewer1 · 2018-11-03
**Limited novelty and unclear motivation**

**Rating:** 5
**Confidence:** 4

**Review:**

The paper addresses the incremental few-shot learning problem where a model starts with base network and then introduces the novel classes, building a connection between novel and base classes via an attention module.

Strengths:
+ clear writing.
+ the experiments are compared with related work and the ablation studies can verify the effectiveness of the proposed (or "introduced" would be a precise term) recurrent BP.

Weakness:

- [Novelty]
The paper title is called attention attractor network, which shares very relevance to previous CVPR work (Gidaris & Komodakis, 2018). So the first thing I was looking for is the clear description of the difference between these two. Unfortunately, in related work, authors mention the CVPR work without stating the difference (last few lines in Section 2). As such, I don't see much novelty in the paper compared with previous work. Eqn. (7)-(10) explicitly describes the attention formula. What's the distinction from the CVPR work?

- [Motivation of the regularizer using Recurrent BP is not clear]
The use of recurrent BP is probably the most distinction from previous work. However, I don't see a clear description on why such a technique is necessary.

Starting from the first line in Section 3.3, "since there is no closed-form of the regularizer in Eqn (13)", E needs BPTT or the introduced recurrent BP. This part is simply a re-adaption of other algorithms. A very simple question is, how about use other regularizers to replace Eqn (13)?

- [Some experiments missing]
The experiments section 4.6 uses a case of None and "best WD" to address some of my concerns. This is good. Does the "gamma random" indicates only E is used without the ||W||^2? why the best WD for one-shot is zero? This implies the model is best for applying no weight decay?

What's the effect of using the recurrent BP technique to the CVPR work? Is there some similar improvement? If yes, then the paper makes some contribution by the regularization. If not, what's the reason?

How about using the truncated BPTT with a larger T?

In general, I think the recurrent BP part should be the highlight of the paper and yet authors fail to spread such a spirit in the abstract or title. And there are some experiments missed as I mentioned above.

---

> ### Author Response · Authors · 2018-11-09
> **Author Response**
>
> Thank you for the review. We are currently revising the paper and will incorporate your helpful suggestions (to add discussion to CVPR work, add BPTT with larger T, and highlight the RBP algorithm).
>
> - Novelty: First, we would like to address the novelty issue. Although both our work and the CVPR paper uses attention mechanism, the two methods are actually very different. The new weights in their method are based on Prototypical Networks, i.e. simply averaging the embedding. We however optimize the weights on the new task and  backprop through the optimization, which is a challenging step in learning our model. The attention mechanism is also formulated differently. Whereas the attended content is used as multiplicative gating in their work, we used it as an additive energy term in the overall objective function to optimize.
>
> - Applying RBP to the CVPR work: The CVPR work is based on a Prototypical Network which computes weights for the novel classes in a single layer, and regular backpropagation is sufficient. Since there is no iterative optimization involved, we do not see anything that allows us to apply RBP to the CVPR work, or any need for it.
>
> - Motivation of RBP: Since we have an iterative optimization procedure in the model, directly differentiating through the procedure is not straightforward. Also, as shown in the experiment, regular backprop through time does not learn a stable objective function. Prior work focus on the case where there is a closed form solution (Bertinetto et al. 2018), where RBP allows us to backprop through any converging optimization layers, which is more general.
>
> - Best WD: Yes, in that experiment, no weight decay is needed (although adding a small amount of WD does not hurt the performance). In the other experiments, we found a small amount of weight decay (1E-5) helped.
>
> - BPTT with larger T: Thank you for the suggestion. We are currently adding more experiments that use a larger T for BPTT and will update the paper with the latest results.

---

> > ### Author Response · Authors · 2018-11-27
> > **Added experiments using T=200**
> >
> > We would like to once again thank the reviewer for the insightful comments. As promised, we have added experiments with T=200 for BPTT baselines (which is 10x longer than the RBP steps). Similar to what we have seen in T=100, we observe a large degradation of performance when solving until convergence. This shows that the introduced RBP algorithm is a modular way to learn energy functions that are less sensitive to how they are minimized in the forward computation.

---

> > ### Comment · AnonReviewer1 · 2018-12-08
> > **Thanks for the rebuttal. But the response is kinda poor.**
> >
> > Dear authors,
> >
> > I read through your feedback and have a quick scan of other reviews. Thanks for your update. I really appreciate. I would keep the score unchanged (5: weak reject).
> >
> > [Novelty]: I don't view the differences you mentioned in my response, alongside with that in other two reviewers (training base data used in CVPR vs not used in yours, etc) as a "very big difference" to call it novelty.
> >
> > [Experiments]: the question as to why not use other regularizers other than the proposed one is not answered. "Starting from the first line in Section 3.3, "since there is no closed-form of the regularizer in Eqn (13)", E needs BPTT or the introduced recurrent BP. This part is simply a re-adaption of other algorithms. A very simple question is, how about use other regularizers to replace Eqn (13)?"
> >
> > [Applying RBP]: What I mean is that your work is: attention with incremental learning + RBP, right? The CVPR work is kinda the first thing only (attention with incremental learning). How about you apply the RBP idea into the CVPR work and see if there is some improvement. They open-sourced the code (not sure tf or pytorch or others) and do need some time to incorporate into their work though.
> >
> > [Motivation of RBP and Organization of the paper] I noticed you changed the related work part about the comparison with CVPR paper. This is good.
> >
> > As regard to the motivation of the proposed regularization, you mentioned, in the new manuscript, that
> > "Since in our settings the model cannot see base class data in each few-shot learning episode, different from
> > Hariharan & Girshick (2017); Wang et al. (2018), it is challenging to jointly classify both base and
> > novel categories using a vanilla logistic regression. Towards this end, we propose to add a learned
> > regularizer, which is learned by differentiating through few-shot learning iterations. "
> >
> > So here is how the logic goes as fas as I understand:
> > challenging to classify both base and novel classes -> a new regularizer is needed -> adopt the RBP (which is not a proposed method but rather you applied existing one).
> >
> > Why is a regularizer needed if learning many classes is challenging? I don't see a strong motivation here.
> >
> > ----
> > Sum-up:
> >
> > 1. As a research paper, novelty is quite limited in the paper even though authors try to explain the difference with previous method.
> >
> > 2. Motivation of RBP part is not clear. This one should be the highlight of your paper and I don't see such a change in the paper organization after rebuttal.
> >
> > 3. Some part (figures, notations, experiments) are not clear. Probably you have already revised them. I don't check this in details though. This negative impression is witnessed from other comments in this forum.

---

### Official Review · AnonReviewer2 · 2018-11-03
**Import discussions missing**

**Rating:** 5
**Confidence:** 3

**Review:**

This paper proposes a novel few-shot learning method that achieves better overall accuracies on base and novel classes. The key idea is to regularize the learning of novel classes such that base classes are not forgotten.

I mainly have the following two concerns.

-In Table 2, I observe that performance on novel classess is actually not improved. The main improvement lies in overall accuracy. As numbers of training samples between base and novel classes are not balanced, there must be some trade-off between  obtaining better performance on base or novel classes. For instance, stopping early when training on novel classes would result in high base accuracy but low novel accuracy. Fine-tuning on novel classes for more iterations would lead to high novel accuracy but  low base accuracy. Such trade-off can be also controlled by simply over-sampling novel or base classes.  I would suggest the authors to study more on understanding this trade-off. In addition, another naive baseline is to train a softmax classifier at the second stage on both base and novel class training samples and sample mini-batch by uniformly sampling over novel and base classes.

-The following two papers extensively studied the problem of achieving better overall accuracies on base and novel classes. Including comparison and discussion with those two papers will enhance this paper further.
Low-Shot Learning from Imaginary Data
low-shot visual recognition by shrinking and hallucinating features

---

> ### Author Response · Authors · 2018-11-09
> **Author Response**
>
> We thank the reviewer for the comments and pointing out related work. We are revising our paper and adding the discussion of these and other relevant papers. In response to one of the public comments, we have compared our approach to these two papers:
>
> The ICCV 2017 paper proposes the SGM loss, which makes the learned classifier from the few-shot examples have a smaller gradient value when learning on all examples. The CVPR 2018 paper proposes the prototypical matching networks, a combination of prototypical network and matching network. The paper also adds hallucination, which generates new examples.
>
> In contrast to these approaches, we directly learn a logistic regression classifier during the few-shot episode, which is very simple and straightforward. Although vanilla logistic regression has been shown to be worse in these prior work (since the logistic regression cannot see old data), we found that it can be improved significantly by differentiating through the few-shot learning iterations, taking into account the additional regularizer..
>
> - Uniform samples: We also would like to emphasize that, in the learning of novel classes, the base class data is *not* available, thus making the problem very challenging. Therefore, the proposed “naive baseline” which samples a mini-batch uniformly over novel and base classes, will not be comparable to the new approach introduced in the paper, which does not rely on reviewing the old data.
>
> - Early stopping: Since we are learning an objective function that needs to be solved until convergence. Stopping early is possible but that relies on an external validation set, which might not be available since we do not have access to the old data when learning the novel classes.
>
> Lastly, the reviewer is right that there is a trade-off between learning novel and remembering old classes. Getting better results on the novel class is is indeed possible but has the undesired effect, of catastrophic forgetting. In our setting of incremental few-shot learning the goal is to have the best performance on *both base and novel classes*. Hence we focus on the \delta bar metric, and our method has a clear win on this crucial metric.

---

> > ### Author Response · Authors · 2018-11-25
> > **Key difference from low-shot learning papers**
> >
> > Once again we thank the reviewer for pointing out the related work on low-shot learning (CVPR18 & ICCV17). We are closely studying them and planning to incorporate their dataset into our experiments. However, we would like to re-emphasize that, one of the key differences between "low-shot learning" and our "incremental few-shot learning" is that "low-shot learning" has access to the training data of base classes during the few-shot learning stage, whereas our "incremental few-shot learning" does not. This makes our problem setup much more challenging, and also more practical since the model does not need to carry the full training data with it. We hope that this addresses the potential misunderstanding.

---

### Official Review · AnonReviewer3 · 2018-11-04
**The problem of incremental few-shot learning is interesting and the presented meta-learning method seems to be effective, but the novelty is limited.**

**Rating:** 5
**Confidence:** 5

**Review:**

This work addresses incremental few-shot learning that learns novel classes without forgetting old classes, which is interesting and different from conventional few-shot learning that considers only the few-shot learning task of interest. This problem is also related closely to the important problem of life-long learning.

This work presents an interesting framework based on meta-learning by learning to learn how to attend to the old classes using an attention mechanism. Experimental results also show improvement over two related works on incremental few-shot learning. The writing is quite clear. Some concerns, especially its novelty, are listed below.

1. The novelty appears to be limited. The presented framework looks quite similar to the recent work

Spyros Gidaris and Nikos Komodakis. Dynamic few-shot visual learning without forgetting. CVPR'18

that addresses the same problem in a similar manner: 1) learn a base feature extractor and classifier; and then 2) attend to old classes also via meta-learning and attention mechanism.
As mentioned by the authors, "The main difference to this work is that we use an iterative optimization to compute W_b". More discussions on the iterative optimization and why it matters may be helpful.

Another related work is "Deep Meta-Learning: Learning to Learn in the Concept Space", Arxiv'18, that also relies on an external base classes for few-shot learning. Similar to the proposed research, it also learns a feature extractor and a classifier from the base classes, which are used to regularize the learning of novel classes, in an end-to-end meta-learning manner. Extending it for the incremental setting seems natural.

2. To learn a few novel classes, all U_k on old classes are relearned, which seems quite time-consuming with a large vocabulary of base classes.

3. To learn a few novel classes, old data on base classes are still required, which seems different from how humans learn -- humans learn novel concepts solely from a few examples without forgetting old concepts, without requiring examples on old concepts.

---

> ### Author Response · Authors · 2018-11-09
> **Author Response**
>
> Thank you for the review. We would like first explain the novelty aspect of our paper.
>
> - Novelty: Although both our work and the CVPR paper use an attention mechanism, the two methods are actually very different. The new weights in their method are based on Prototypical Nets, i.e. simply averaging the embedding. We however optimize the weights on the new task and backprop through the optimization, which is a challenging step in learning our model. The attention mechanism is also formulated differently. Whereas the attended content is used as multiplicative gating in their work, we used it as an additive energy term in the overall objective function to optimize.
>
> Secondly, there seem to be a couple crucial misunderstandings in the review. We will revise our paper to make sure that our points are clearly stated.
>
> - Learning of novel classes needs old data: We are afraid that there might be a big misunderstanding. The whole incremental few-shot learning problem is set up so that reviewing old data is *not* allowed. Otherwise the problem can be very trivial to solve: just sample some old data and new data and train jointly. We believe that learning novel classes *without* reviewing old data is an important and challenging problem, especially learning it iteratively, since many models will run into catastrophic forgetting. We have shown that while BPTT does not perform well in this scenario, the proposed meta-learning algorithm can solve it.
>
> - Learning of novel classes involves relearning U_k. During learning of novel classes, U_k is fixed and *not* re-learned. U_k is learned during the meta-learning stage, where the novel classes are subsampled from the training set classes (Train_B set). Also the size of U_k is the same as a fully connected softmax layer, which is quite small compared to all the parameters of a deep CNN model.

---

> > ### Comment · AnonReviewer3 · 2018-12-08
> > **More insights may help to improve the novelty**
> >
> > Thanks for the clarification. There is no misunderstanding during the original review.
> >
> > The meta-learning happens in the second stage of the framework, and it involves the data in the first stage, as the query Q_{a+b} contains old classes, quoted below
> >
> > "During meta-learning, E are updated to minimize an expected loss of the query set
> > Qa+b which contains both old and new classes, averaging over all few-shot learning episodes."
> >
> > So an interesting extension would be to study whether it is also possible to use only the learned feature extractor and classifier from the first stage.
> >
> > This framework is quite interesting but it appears incremental given Gidaris & Komodakis, CVPR'18, though several modifications are proposed. An interesting extension would be to provide new insights into the framework, say, on how it attends to old classes, when it would fail, and on the assumption of the relatedness of old data (D_a) and new data (D_b).

---

### Public Comment · (anonymous) · 2018-10-24
**Interesting work, questions about problem setting and optimizer**


The idea of incremental few-shot learning in this paper is quite interesting. After reading the paper, I have two questions detailed in the below.

Q1. The same problem has been proposed and studied in two recent papers “Low-shot Visual Recognition by Shrinking and Hallucinating Features (ICCV 2017)” and “Low-Shot Learning from Imaginary Data (CVPR 2018)”. They address the same problem of classifying novel classes with a few labeled examples based on identifying a set of base classes. They also categorize the classes as “base” and “novel” class as this paper does. Since I did not find any discussion about these two papers in this paper, can you provide some comments about their differences?

Q2. In this paper, you use recurrent back-propagation as an optimizer, but most previous few-shot learning methods use SGD. Recurrent back-propagation is widely used in NLP because of the sequential nature of texts. However, an image is rarely treated as a sequence. Is there any particular reason for using recurrent back-propagation? Or did you see any critical advantages of using recurrent back-propagation rather than using SGD?

I am looking forward to your reply. Thanks a lot!

---

> ### Author Response · Authors · 2018-10-28
> **Thank you for the comment**
>
> 1) Thank you for your comments. We will add the discussion in our next version of the paper. Note that in our paper we compared to LwoF, which has better performance than the two papers mentioned above. We are planning to add experiments using the dataset proposed by Bharath & Girshick for more thorough comparison.
>
> The ICCV 2017 paper proposes the SGM loss, which makes the learned classifier from the few-shot examples have a smaller gradient value when learning on all examples.
>
> The CVPR 2018 paper proposes the prototypical matching networks, a combination of prototypical network and matching network. The paper also adds hallucination, which generates new examples.
>
> Different from these approaches, we directly learn a logistic regression classifier during the few-shot episode, which is very simple and straightforward. Although it has been shown to be worse in these prior work, we found that it can be improved significantly by backprop through the few-shot learning iterations to learn additional regularizer terms.
>
>
> 2) We think you might be mixing back-propagation through time (BPTT) commonly used to train recurrent neural networks with recurrent back-propagation (RBP). We are not trying to replace the SGD algorithm, but just proposing to use RBP to take the gradients. Typically, when training RNNs, people use backpropagation through time (BPTT), which unrolls the computation graph and takes the gradient. RBP is a different way of taking gradients, if the recurrent process converges to a fixed point. Here we found RBP is a better tool for learning the energy functions.

---

### Comment · AnonReviewer1 · 2018-10-27
**Some clarification**

Hi,

I have some confusions in the paper.

(1) what's h_tilde in Eqn. (1)?
(2) how \delta_bar is computed in Table 1? for example, "LwoF (our implementation)", is it supposed to be (56.97 + 52.37)/2 - 74.58 = -19.91?
(3) Fig.1, the iterative process corresponds to the M-loop in Alg. 1? If so, it seems that the M-loop deals with L_Q, which is the query set, the "iterative solver" in Fig. 1 deals with support set only.

---

> ### Author Response · Authors · 2018-10-28
> **Response**
>
> Thank you for the questions.
>
> (1) h_tilde is the original hidden representation, and we augment it with an extra dimension with value=1.
>
> (2) When jointly testing base and novel classes, we quantify the drop in performance in each category, relative to testing the base and novel classes separately as follows:
>
> If the Acc_A is base accuracy, Acc_B is few-shot accuracy, and Acc_joint is joint accuracy. Within Acc_joint, Acc_A’ is the base accuracy when tested jointly, and Acc_B’ is the few-shot accuracy when tested jointly. Then, \delta_bar is computed as:
>
> \delta_bar = ½ (Acc_A’ - Acc_A) + ½ (Acc_B’ - Acc_B)
>
> (3) The iterative process corresponds to Line 5 in Alg. 1, where it solves the L_S loss. M-loop is the backpropagation of gradients of the loop.

---

### Public Comment · (anonymous) · 2018-11-05
**Question about the details**

Hi, I have the following confusion: How is the attractor $U_k$ for the base class which is stored in the knowledge base generated? Apologies if I missed something important.

Looking forward to your reply. Thank you.

---

> ### Author Response · Authors · 2018-11-05
> **Response**
>
> Hi, $U_k$ are learned as slow weights in the meta-training. Thanks!

---

### Meta-Review · Area_Chair1 · 2018-12-14

**Confidence:** 4
**Recommendation:** Reject

**Metareview:**

This paper proposes an approach for incremental learning of new classes using meta-learning.
Strengths: The framework is interesting. The reviewers agree that the paper is well-written and clear. The experiments include comparisons to prior work, and the ablation studies are useful for judging the performance of the method.
Weaknesses: The paper does not provide significant insights over Gidaris & Komodakis '18. Reviewer 1 was also concerned that the motivation for RBP is not entirely clear.
Overall, the reviewers found that the strengths did not outweigh the weaknesses. Hence, I recommend reject.